# Analysis of Viral Promoters for Transgene Expression and of the Effect of 5′-UTRs on Alternative Translational Start Sites in *Chlamydomonas*

**DOI:** 10.3390/genes14040948

**Published:** 2023-04-21

**Authors:** Justus Niemeyer, Laura Fischer, Frank O’Neill Aylward, Michael Schroda

**Affiliations:** 1Molecular Biotechnology & Systems Biology, RPTU Kaiserslautern-Landau, Paul-Ehrlich-Straße 23, 67663 Kaiserslautern, Germany; niemeyej@rhrk.uni-kl.de (J.N.); lafische@rptu.de (L.F.); 2Department of Biological Sciences, Virginia Tech, Blacksburg, VA 24061, USA; faylward@vt.edu

**Keywords:** synthetic biology, golden gate cloning, giant virus, viral promoters, mCherry isoforms, 5′-UTR, microalgae, *Chlamydomonas reinhardtii*

## Abstract

Microalgae biotechnology has the potential to produce high quality bioproducts in a sustainable manner. Here, *Chlamydomonas reinhardtii* has shown great potential as a host for biotechnological exploitation. However, low expression of nuclear transgenes is still a problem and needs to be optimized. In many model organisms, viral promoters are used to drive transgene expression at high levels. However, no viruses are known to infect *Chlamydomonas*, and known viral promoters are not functional. Recently, two different lineages of giant viruses were identified in the genomes of *Chlamydomonas reinhardtii* field isolates. In this work, we tested six potentially strong promoters from these viral genomes for their ability to drive transgene expression in *Chlamydomonas*. We used *ble*, *NanoLUC*, and *mCherry* as reporter genes, and three native benchmark promoters as controls. None of the viral promoters drove expression of any reporter gene beyond background. During our study, we found that mCherry variants are produced by alternative in-frame translational start sites in *Chlamydomonas*. We show that this problem can be overcome by mutating the responsible methionine codons to codons for leucine and by using the 5′-UTR of *βTUB2* instead of the 5′-UTRs of *PSAD* or *RBCS2*. Apparently, the *βTUB2* 5′-UTR promotes the use of the first start codon. This could be mediated by the formation of a stem-loop between sequences of the *βTUB2* 5′-UTR and sequences downstream of the first AUG in the *mCherry* reporter, potentially increasing the dwell time of the scanning 40S subunit on the first AUG and thus decreasing the probability of leaky scanning.

## 1. Introduction

The unicellular green alga *Chlamydomonas reinhardtii* (*Chlamydomonas*) is gaining increasing interest as a eukaryotic host to produce high value products and biopharmaceuticals. *Chlamydomonas* produces recombinant proteins well, folds them correctly, modifies them post-translationally and can secrete them [1,2,3,4,5,6]. *Chlamydomonas* grows both photoautotrophically and heterotrophically in simple and inexpensive media, which can be scaled up in photobioreactors, making this organism particularly attractive for commercial applications [7]. One limitation of *Chlamydomonas* is the low expression of nuclear transgenes. Several approaches have helped to overcome this problem, such as the use of optimal codon usage, the use of promoters capable of counteracting gene silencing, the integration of introns into the coding sequence and the use of expression strains with defects in transgene silencing [8,9,10,11]. In *Chlamydomonas,* native promoters are commonly used to drive transgene expression [12]. The most frequently used ones are the *PSAD* promoter and the chimeric *HSP70A-RBCS2* promoter (*AR*) [13,14]. Recently, a new synthetic promoter was introduced, the *AβSAP(i)* promoter. This promoter is derived from a fusion of the *HSP70A* and *βTUB2* promoters and harbors synthetic *cis* motifs as well as the *βTUB2* 5′-UTR, into which the first *RBCS2* intron was integrated [2]. In many expression hosts, viral promoters are used to mediate high-level expression of transgenes. Until recently, there were no reports about viruses infecting *Chlamydomonas* and therefore promoters of viruses infecting other organisms were tested. These promoters showed at most weak activity in *Chlamydomonas* [15,16,17,18,19,20]. Recently, viral DNA elements integrated into the genomes of several field isolates of *Chlamydomonas* were identified [21]. These originate from viruses of the orders *Algavirales* and *Imitervirales*, which belong to the so-called giant viruses of the phylum *Nucleocytoviricota*. The *Nucleocytoviricota* have ds-DNA genomes with sizes that can exceed 500 kb and contain 400 to 2500 genes [22,23]. The giant viruses either replicate in the host cytoplasm in virus factories or potentially enter a lysogenic cycle. The segments of the viral genomes found in *Chlamydomonas* are called Giant Endogenous Viral Elements (GEVE) and are several hundred kilobases in length, harboring most of the hallmark genes and thus indicating that near-complete genomes are present [21,24,25]. The GEVEs contain introns and the GC content is about 60%, i.e., similar to that of *Chlamydomonas* at 64% [21,26]. The 1.2-Mb genome of the *Imitervirales* contains about 910 genes, including those encoding enzymes of various DNA repair mechanisms, replication and transcription, and metabolic processes as well as genes encoding proteins of the virion. Unlike the *Imitervirales*, the *Algavirales* do not encode an endogenous RNA polymerase and use that of the host. Some of the most common proteins are the major capsid proteins (MCPs), suggesting a high expression level of the genes encoding them and thus genes with strong promoters [21].

*Chlamydomonas* is the host of at least two distinct lineages of giant viruses. In this study, we tested six promoters of genes encoding capsomeres and RNA polymerases from these GEVEs for their capability to drive transgene expression in *Chlamydomonas* and compared them with three native benchmark promoters. Unfortunately, none of the viral promoters was able to drive the expression of any of the three tested reporter genes. During our studies, we could show that shorter isoforms of mCherry are produced from alternative translational start sites and that this can be overcome by mutating the methionine codons or by using different 5′-UTRs.

## 2. Methods

### 2.1. Strains and Culture Conditions

Chemically competent *E. coli* TOP10 cells were used for heat-shock-mediated transformation of plasmid DNA. Selection was done on LB agar plates containing either 100 µg/mL spectinomycin (Level 0) or 50 µg/mL kanamycin (Level 2). The *Chlamydomonas* UVM4 strain [27] was grown in Tris-Acetate-Phosphate (TAP) medium [28] at a continuous light intensity of ~40 µmol photons m^−2^ s^−1^.

### 2.2. Nuclear Transformation of Chlamydomonas

Transformation was done using glass beads [29]. For this, 10^8^ cells were mixed with 1 µg of linearized plasmid DNA, vortexed for 15 s and carefully spread on agar plates containing either 2 µg/mL zeocin (*ble* modules) or 100 µg/mL spectinomycin (*NanoLUC* and *mCherry* devices). After overnight incubation in the dark, plates were transferred to ~40 µmol photons m^−2^ s^−1^ and zeocin resistant colonies were counted after 8 days.

### 2.3. Cloning

Promoter sequences from two *Algavirales* Capsid-Protein genes (*AVCP1* and *AVCP2*) and from the large subunit of the *DNA-Polymerase B* gene (*AVDPol)* as well as two promoters from *Imitervirales* Capsid-Protein genes (*IVCP1* and *IVCP2*) and the gene encoding the large subunit of the *RNA-Polymerase* (*IVRPol*) were selected from Giant Endogenous Viral Elements [21]. Promoter sequences included sequences ranging from 485 to 519 bp upstream from the translational start sites (Appendix A). All sequences were synthesized as gBlocks (IDT) with flanking *BbsI* recognition sites creating GGAG and AATG fusion sites according to the MoClo syntax for plant genes [30,31]. Each promoter sequence was combined with the pICH41295 vector [31] and assembled in a ligation-restriction reaction with *BbsI* and T4 DNA ligase (level 0) giving rise to pMBS978 (*AVCP1*), pMBS979 (*AVCP2*), pMBS980 (*AVDPol*), pMBS981 (*IVCP1*), pMBS982 (*IVCP2*) and pMBS983 (*IVRPol*). The level 1 *ble* modules were assembled into destination vector pICH47732 [31] by combining the following parts: A1-B2_pCM0-016 (*PSAD* promoter+5′-UTR), A1-B2_pMBS96 (*AR* promoter+5′-UTR), A1-B2 *AβSAP(i)* promoter+5′-UTR(i) [2], A1-B2_pMBS978-pMBS983, B3-B5_pCM0-077 (*ble(i)* CDS) and B6-C1_pCM0-119 (*RPL23 3′-UTR*). Level 2 devices for *mCherry* and *NanoLUC* reporters were generated by assembling the same parts as well as B3-B4 pCM0-063 (*NanoLUC* CDS), B3-B4_pCM0-067 (*mCherry(i)* CDS), and B5_pCM0-100 (*3xHA* CDS) into destination vector pMBS807 [32] conferring resistance to spectinomycin after transformation into *Chlamydomonas*.

Mutated *mCherry(i)* reporters were generated by site-directed mutagenesis using B3-B4_pCM0-067 (*mCherry(i)* CDS) as a template. For substituting the methionine codon with a leucine codon, the following oligonucleotides with flanking *BbsI* sites (bold) were used: M10L-for: TTGAAGACAAcTcGCCATCATCAAGGAGTTCATGCG and M10L-rev: TTGAAGACAACgAgGTTATCCTCCTCGCCCTTGCT, M17L-for: TTGAAGACAAcTcCGCTTCAAGGTGCACATG and M17L-rev: TTGAAGACAAGgAgGAACTCCTTGATGATGGCCATG, M23L-for: TTGAAGACAAcTcGAGGGCTCCGTGAACG and M23L-rev: TTGAAGACAACgAgGTGCACCTTGAAGCGCAT. PCR products were purified using the Machery–Nagel clean up kit following the manufacturer’s instructions. Purified PCR products were digested with *BbsI* and ligated to create level 0 parts. The sequences for B3-B4_pMBS1033 (*mCherry(i)-M10L* CDS), B3-B4_pMBS1034 (*mCherry(i)-M17L* CDS) and B3-B4_pMBS1035 (*mCherry(i)-M23L* CDS) were verified by Sanger sequencing. To exchange the 5′-UTRs of the *PSAD* and *AR* promoters with the *βTUB2* 5′-UTR, level 0 parts A1-A3_pCM0-001 (*PSAD* promoter), A1-A3_pCM0-002 (*AR* promoter) and B1-B2_pCM0-033 (*βTUB2* 5′-UTR) were used.

### 2.4. Determination of NanoLUC Activity

All spectinomycin-resistant colonies containing the *NanoLUC* reporter of one transformation event were pooled and grown to mid-logarithmic phase. A total of 1.5 × 10^7^ cells were harvested and resuspended in 3 mL fresh TAP medium yielding a suspension with 5 × 10^6^ cells/mL. From this suspension, 50 µL (2.5 × 10^5^ cells) were transferred into a white 96-well plate. Finally, 50 µL Nano-Glo in its buffer (Promega) were added and NanoLUC activity was measured with a FLUOstar Omega plate reader (BMG Labtech) at 460 nm. Data were analyzed by subtracting the background signal from the recipient strain (UVM4) and normalizing to the bioluminescence signal from the *PSAD-NanoLUC* reporter.

### 2.5. Determination of mCherry Activity

All spectinomycin-resistant colonies containing the *mCherry* reporter of one transformation event were pooled and grown to mid-logarithmic phase. Then, 10^7^ cells were harvested and resuspended in 200 µL of a 100 mM MES/Tris buffer, pH 7.0. The 10^7^ cells were transferred into a black/clear bottom 96-well plate. Cells were centrifuged for 2 min at 30× *g* and 25 °C, and fluorescence signals were measured with a FLUOstar Omega plate reader (BMG Labtech) using 584 nm excitation and 620 nm emission filters. Data were analyzed by subtracting the background signal from the recipient strain (UVM4) and normalizing to the fluorescence signal from the *PSAD-mCherry(i)* reporter.

### 2.6. Protein Analyses

Whole-cell protein extraction, SDS-PAGE, semi-dry blotting and immunodetections were performed as described previously [32]. Total cell proteins corresponding to 1 µg chlorophyll were used [33]. For immunodetection, a mouse primary antibody against the HA epitope (Sigma, Darmstadt, Germany, 1:10,000) and a secondary antibody (m-IgGκ BP-HRP, Santa Cruz Biotechnology, Dallas, TX, USA, 1:10,000) were used.

## 3. Results

### 3.1. Viral Promoters Are Not Functional in Chlamydomonas

We aimed at testing promoters from Giant Endogenous Viral Elements (GEVE) [21] for their ability to drive transgene expression in Chlamydomonas. Since viral capsid genes are generally strongly expressed, we chose two promoter sequences from Algavirales major capsid protein genes (AVCP1 and AVCP2) and two from Imitervirales major capsid protein genes (IVCP1 and IVCP2) from the GEVEs. Since members of the Imitervirales encode their own RNA polymerase, the promoters of these viruses might not be recognized by the host RNA polymerase but by that of the virus. Since the promoter of the viral RNA polymerase gene must be recognized by the host RNA polymerase, we included the promoter of the large subunit of the RNA polymerase (IVRPol) from the *Imitervirales* GEVE in our study. Since *Algavirales* do not encode an RNA polymerase, we included the promoter of the large subunit of the DNA polymerase B (AVDPol) for comparison. The promoter sequences comprised sequences from 485 to 519 bp upstream of the translational start codon (Figure 1 and Appendix A).

The six promoter sequences were synthesized as level 0 parts following the syntax of the *Chlamydomonas* MoClo kit [30]. The resulting viral promoters as well as benchmark promoters *PSAD*, *AR* and *AβSAP(i)* were combined with one of the following three reporter genes: (1) The antibiotic resistance marker *ble*, which encodes the *Sh*Ble protein. The latter binds and inactivates Bleomycin which causes DNA double strand breaks. The resistance level to Bleomycin correlates with the expression level of the *Sh*Ble protein, thus allowing a fast and simple read-out of promoter activity [34,35]. (2) Nanoluciferase (NanoLUC), which displays high sensitivity with sustained luminescence as well as a strong linearity between luminescence and protein quantity [30,36]. (3) The red monomeric fluorescent protein mCherry [37] (Figure 1). Notice that reporter genes *ble* and *mCherry* contain the first *RBCS2* intron *(i)*, whereas *NanoLUC* contains no intron. We added a sequence encoding a triple-HA tag to the *NanoLUC* and *mCherry(i)* reporter genes for detection on Western blots. The *RPL23* terminator was used in all constructs, as it confers constant expression levels with different transgene combinations [2]. Vectors with *NanoLUC* and *mCherry(i)* reporters additionally contained the *aadA* cassette conferring resistance to spectinomycin. In total, 27 reporter constructs were assembled (Figure 1) and transformed into the *Chlamydomonas* UVM4 expression strain [27].

Each of the *ble(i)* reporter constructs was transformed six times, and zeocin-resistant colony-forming units (CFU) were counted and normalized to the numbers obtained with the *PSAD-ble(i)* reporter. Normalized counts revealed that, in contrast to the benchmark promoters, none of the viral promoters drove *ble* expression to levels high enough to confer resistance to zeocin above background (Figure 2A). Of the benchmark promoters, the *AβSAP(i)* promoter performed significantly better than the *PSAD* promoter, while differences between *AR* and the other promoters were not significant.

Transformation of the *NanoLUC* reporter constructs resulted in approximately 1200 spectinomycin-resistant colonies each, since expression of the *aadA* marker was independent of the promoter used to drive the *NanoLUC* reporter. To average out position effects for the *NanoLUC* reporter, all colonies of an individual transformation event were pooled [11,38]. Three transformant pools were generated for each construct, and each pool was further divided into two replicates before NanoLUC luminescence was measured. Signals were normalized to those obtained for the *PSAD-NanoLUC* reporter. As was observed for the *ble* reporter, none of the viral promoters drove *NanoLUC* expression to levels high enough to produce luminescence above background (Figure 2B). Of the benchmark promoters, the *AβSAP(i)* promoter performed significantly better than the *AR* promoter, which again performed significantly better than the *PSAD* promoter.

As for the *NanoLUC* reporter constructs, approximately 1200 spectinomycin-resistant transformants generated with each *mCherry(i)* reporter construct were pooled. Pools from three independent transformation events were further divided into three replicates before mCherry fluorescence was measured. Signals were normalized to those obtained for the *PSAD-mCherry(i)* reporter. As we observed for the *ble* and *NanoLUC* reporters, none of the viral promoters drove *mCherry* expression to levels high enough to produce fluorescence above background (Figure 2C). Of the benchmark promoters, the *AβSAP(i)* promoter performed significantly better than the *AR* and *PSAD* promoters, between which no significant differences were observed.

### 3.2. The βTUB2 5′-UTR Prevents the Production of a Shorter mCherry Isoform

To verify that the bioluminescence and fluorescence signals obtained for the transformant pools correspond with reporter protein abundance, we made use of the 3× HA tag fused to each reporter and analyzed their abundance conferred by the benchmark promoters via Western blotting. As shown in Figure 2D, the abundance of the reporter proteins correlated well with the respective bioluminescence and fluorescence signals. However, for mCherry, we noticed a less intense protein band just below the expected band in transformant pools generated with *PSAD-* and *AR-mCherry(i)* reporters, but not in pools generated with the *AβSAP(i)-mCherry(i)* reporter (Figure 2D, arrowhead).

The mCherry CDS contains three additional in-frame codons for methionine that are located 10, 17 and 23 codons downstream of the regular start codon (Figure 3A). While mCherry isoforms produced from M17 and M23 are non-functional, the variant produced from M10 results in a functional fluorescent protein [39]. Despite the fundamentally different mechanisms of translation initiation in pro- and eukaryotes, mCherry variants deriving from internal translational start sites are produced from reporter constructs in *E. coli, Mycobacterium tuberculosis* and *Saccharomyces cerevisiae* [39,40]. To test whether the additional protein band in transformants harboring the *PSAD-* and *AR-mCherry* reporters derived from variants produced from alternative translational start sites, we mutated the M10, M17 and M23 codons in our reporter constructs to codons for leucine. As shown in Figure 3B, the smaller protein band (arrowhead a) vanished in M10L mutants in both *PSAD-* and *AR-mCherry(i)* reporters, but not in M17L or M23L mutants. However, another protein band with an even smaller protein occurred in M10L mutants (arrowhead b). This protein band was more prominent in transformant pools harboring the *PSAD-mCherry(i)* reporter than in transformant pools harboring the *AR-mCherry(i)* reporter. We assume that it results from alternative translation initiation at M17 and would therefore not give rise to functional mCherry [39].

So far, our data indicate that the smaller proteins derive from internal translational start sites. However, why do they occur only with the *PSAD-* and *AR-mCherry(i)* reporters but not with the *AβSAP(i)-mCherry(i)* reporter? We hypothesized that the *βTUB2* 5′-UTR in the *AβSAP(i)-mCherry(i)* reporter construct disfavors the use of translational start codons downstream of the first AUG, while the use of alternative start codons is favored by the *PSAD* and *RBCS2* 5′-UTRs. To test this idea, we replaced the *PSAD* and *RBCS2* 5′-UTRs in the *PSAD-* and *AR-mCherry(i)* reporters with the *βTUB2* 5′-UTR. As shown in Figure 3B, this step indeed abolished the use of internal translational start sites in both reporters.

Finally, we measured mCherry fluorescence in transformant pools generated with the engineered reporter constructs to reveal adverse effects of the engineering steps on reporter performance. Most strikingly, we found strongly reduced mCherry fluorescence in transformant pools producing the M17L mCherry variant (Figure 3C), suggesting that not only the segment between M10 and M17 but also M17 itself is necessary for mCherry functionality [39,40]. All other engineering steps had no significant effect on mCherry fluorescence.

## 4. Discussion

In contrast to three benchmark native *Chlamydomonas* promoters, none of the six viral promoters tested was able to drive expression of three different reporter genes (Figure 2A–C). Possibly, the viral promoter sequences suffered from mutations rendering them inactive as a consequence of selective pressure against the expression of viral genes after endogenization of the viral genome. Alternatively, specific transcription factors encoded on the viral genome might be required for making most of the viral promoters accessible for transcription. We may also have missed enhancer sequences present up- or downstream of the promoter sequences selected. To understand why the six promoters were not functional, giant viruses able to infect *Chlamydomonas* must be identified to study viral gene expression during infection.

When comparing the activity of the three native benchmark promoters, the *AβSAP(i)* promoter drove the expression of three and two reporter genes to significantly higher levels than the *PSAD* and *AR* promoter, respectively (Figure 2). Notice, however, that the *AβSAP(i)* promoter contains the first *RBSC2* intron in its 5′-UTR [2], while the 5′-UTRs of the two other promoters do not contain an intron. It is known that the presence of regularly spaced introns with appropriate exon boundaries enhances transgene expression, presumably via a process termed intron-mediated enhancement [8,12,41,42]. Hence, it is not clear to what extent the better performance of the *AβSAP(i)* promoter is due to the additional intron. In any case, the *AβSAP(i)* promoter represents a functional unit with superior performance on various reporter genes containing (*mCherry*, *ble*) or lacking (*NanoLUC*) introns.

By employing the mCherry reporter gene with four in-frame AUGs within the first 69 nt of its coding sequence, we could observe a frequent use of the second AUG in the context of the *PSAD* and *RBCS2* 5′-UTRs, but not in the context of the *βTUB2* 5′-UTR, where only the first AUG was used (Figure 2D and Figure 3B). Mutation of the second AUG to CUC (Leu) in the context of the *PSAD* and *RBCS2* 5′-UTRs led to a more frequent use of the third AUG (Figure 3B). Apparently, the *βTUB2* 5′-UTR promotes the use of the first AUG much better than the *PSAD* and *RBCS2* 5′-UTRs. This can be for several reasons [43]: first, the first AUG may be skipped if the surrounding sequence is not adhering to the “Kozak consensus” sequence [44], which in *Chlamydomonas* is 5′-(A/G)(A/C)(C/A)AUGG-3′ [45]. Since this consensus sequence is met by all three 5′-UTRs, and sequences are the same in the *RBCS2* and *βTUB2* 5′-UTRs, leaky scanning by an unfavorable “Kozak consensus” appears unlikely. Second, leaky scanning is promoted if the length of the 5′-UTR is <~20 nt [46]. The 5′-UTRs of *PSAD*, *RBCS2* and *βTUB2* comprise 43 nt, 23 nt, and 120 nt, respectively. Hence, the short length of the *RBCS2* 5′-UTRs could promote leaky scanning. Since RBCS2 is one of the most abundantly produced proteins in *Chlamydomonas*, inefficient translation initiation by a too short 5′-UTR of *RBCS2* also seems unlikely. Third, initiation efficiency can be influenced by secondary structures in the 5′-UTR. A strong stem-loop structure just downstream of the first AUG will stall the scanning 40S subunit, increasing its “dwell time” on the first AUG, and thus decreasing the probability of leaky scanning [47]. Indeed, the RNAfold program of the Vienna RNA package [48] predicted sequences between the first and second AUG to form a stable stem-loop structure with sequences from the *βTUB2* 5′-UTR, but not with sequences from the *PSAD* and *RBCS2* 5′-UTRs (Appendix A).

Leaky scanning can have several negative effects. First, frequent “missing” of the first AUG may lead to inefficient translation of a transcript and thus to low yields of the encoded protein. In the case of the mCherry reporter, the leaky scanning problem appears to be overcome by the production of a functional protein from the second AUG (M10), thus preserving its function as a reporter for promoter activity. The fluorescence signals produced by the constructs with *PSAD* and *RBCS2* 5′-UTRs (where the M10 variant occurs) are not different from the signals produced by the corresponding constructs with *βTUB2* 5′-UTRs (where the M10 variant does not occur) (Figure 3C). Hence, the M10 mCherry variant seems to have a similar activity as the full-length protein. Another negative effect of leaky scanning concerns fluorescent reporter proteins where a targeting peptide is fused N-terminally to the reporter. In this case, the use of a second, in-frame AUG can result in the production of a reporter lacking the targeting peptide, thus leading to erroneous localization [39].

While replacing the *PSAD* and *RBCS2* 5′-UTRs by the *βTUB2* 5′-UTR solved the leaky scanning problem for the *mCherry* reporter (Figure 3B), other transgenes may require different 5′-UTRs to avoid this problem. Here, we would like to point out that the MoClo tool kit facilitates easy exchange of 5′-UTRs since core promoter sequences and 5′-UTRs are available as independent genetic parts [30]. It might also be an option to design transgenes to contain a 5'-UTR with a propensity to form a stem-loop structure immediately downstream of the AUG start codon.

## Figures and Tables

**Figure 1 genes-14-00948-f001:**
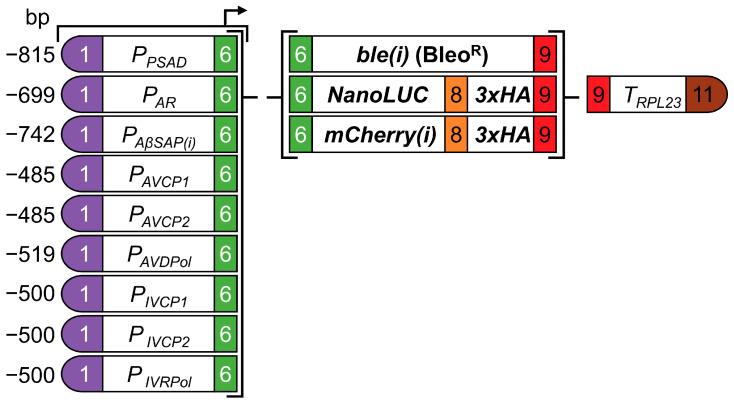
Schematic representation of reporter modules to test viral promoter activity in *Chlamydomonas.* The domesticated viral promoter sequences of capsomer (CP) and DNA/RNA-polymerase genes (D/RPol) were synthesized according to the MoClo syntax, generating fusion sites suitable for cloning into positions A1-B2. The color code for the 11 fusion sites was introduced previously [30]. The established promoters *PSAD*, *AR* and *AβSAP(i)* were used as benchmarks. Every promoter was assembled with *ble(i)*, *NanoLUC*, and *mCherry(i)* reporters and the *RPL23* terminator. *NanoLUC* and *mCherry(i)* reporters also contain sequences coding for a *3xHA-tag. (i)* indicates the presence of the first *RBCS2* intron.

**Figure 2 genes-14-00948-f002:**
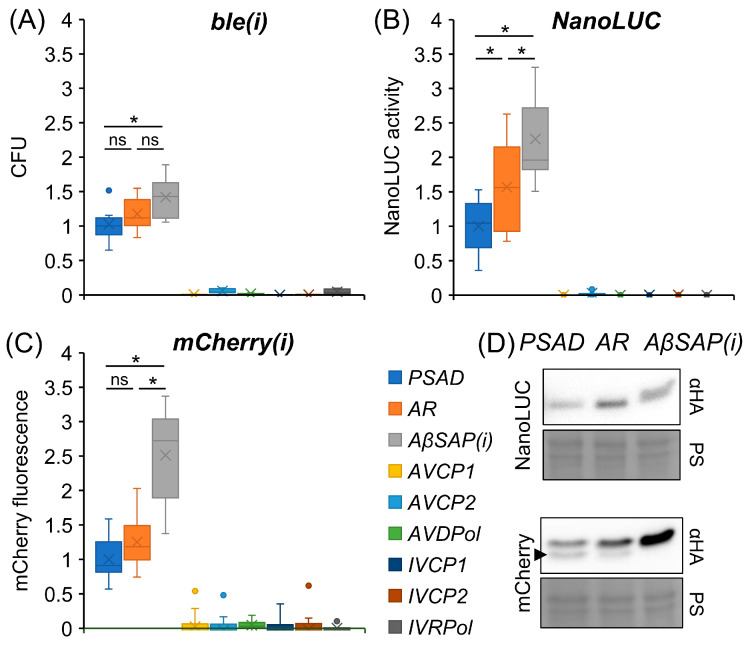
Quantification of viral promoter strength compared to benchmark promoter strength using *ble(i)*, *NanoLUC* and *mCherry(i)* reporters. (**A**) Number of zeocin-resistant colony-forming units (CFU) normalized to CFUs obtained with the *PSAD-ble(i)* reporter (set to 1). After transformation and plating on agar plates containing 2 µg/mL zeocin, CFUs were counted after eight days (*n* = 6). (**B**) Bioluminescence signals from NanoLUC measured in transformant pools normalized to signals obtained for pools generated with the *PSAD-NanoLUC* reporter construct (*n* ≥ 6). (**C**) mCherry fluorescence measured in transformant pools normalized to pools generated with the *PSAD-mCherry(i)* reporter (*n* ≥ 9). (**D**) Comparison of NanoLUC and mCherry protein accumulation in transformant pools. Total cell proteins corresponding to 1 µg chlorophyll were separated by SDS-PAGE and analyzed by immunoblotting using an HA antibody. Ponceau S (PS) staining shows equal loading. The arrowhead points to a shorter mCherry variant. To test for significant differences between *AR*- and *AβSAP(i)-* versus *PSAD* promoter-driven reporters, a one-factor ANOVA was performed using Bonferroni’s multiple comparison test (*: *p* < 0.05; ns: not significant). There were no significant differences between the viral promoters. Dots outside the whiskers represent outliers.

**Figure 3 genes-14-00948-f003:**
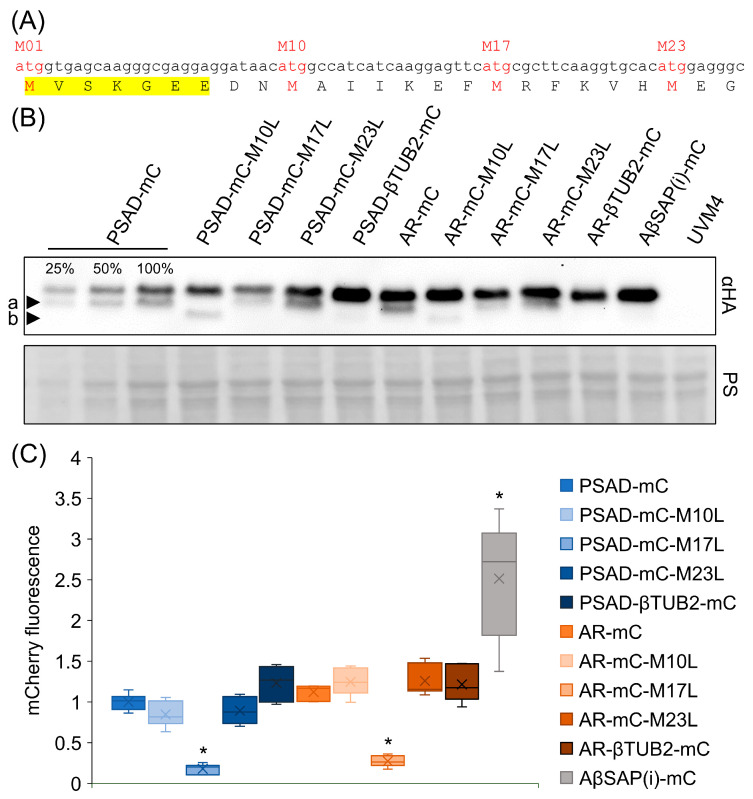
Comparison of the effect of mutated alternative translational start sites and of different 5′-UTRs on the production of mCherry variants. (**A**) Sequence of the 25 N-terminal amino acids of mCherry and the underlying codons. Codons for methionines serving as alternative translation initiation sites are in red letters (M1, M10, M17, M23). The N-terminal fragment derived from eGFP is highlighted in yellow. (**B**) Comparison of mCherry (mC) isoform accumulation in transformant pools. Total cell proteins corresponding to 1 µg chlorophyll were separated by SDS-PAGE and analyzed by immunoblotting using an HA antibody. Alternative translational start codons (M10, M17 or M23) were substituted with codons for leucine (L) in *PSAD-* and *AR-mCherry(i)* reporters. In addition, the 5′-UTRs of *PSAD* and *RBCS2* were replaced by that of *βTUB2*. Transformant pools containing *AβSAP(i)-mCherry(i)* and the UVM4 recipient strain were loaded as controls. Ponceau S (PS) staining shows equal loading. Arrowheads a and b point to shorter mCherry isoforms. (**C**) mCherry fluorescence measured in transformant pools normalized to pools generated with the *PSAD-mCherry(i)* reporter construct (*n* = 6). The *AβSAP(i)-mCherry(i)* fluorescence data from Figure 2C were included for comparison. For significance testing, a one-factor ANOVA was performed using Dunnett’s multiple comparison test against *PSAD-mC* (*: *p* < 0.05). Non-significant tests are not shown.

## Data Availability

The data presented in this study are available in this article and the Appendix A.

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
