# Peer review of "Analysis of Viral Promoters for Transgene Expression and of the Effect of 5′-UTRs on Alternative Translational Start Sites in Chlamydomonas"

_genes, 2023, doi:10.3390/genes14040948_

Round 1

Reviewer 1 Report

In this well written article, Nimeyer and colleagues provide interesting evidence regarding two types of genetic designs in Chlamydomonas reinhartii.

First, they characterized putative virus promoters from virus sequences found in the nuclear genome of C. reinhartii. They found that the tested sequences were not able to drive transgene expression of 3 different reporters. Thanks to their controls, they confirmed previously publishes data (Einhaus et al. 2021. Ref. 2 of the manuscript).

Second, they identified by serendipity a flawn in the widely used fluorescent protein mCherry. They found alternative translation start codons of which one revealed to be a possible bias for in vivo analyses. The authors found that this occurs only with the PSAD and AR promoters-5'UTR parts and that the replacement of the associated 5'UTR removes these alternative translation initiation.

Altogether, the results provided are sound and the experiments well executed. 

I have only minor comments.

1)I found a small mistake in the material section:

Line 101 "B2-B5_pCM0-077 (ble(i) CDS)". The authors probably meant "B3-B5", otherwise the assembly described is impossible.

2)The authors postulate that the 5'UTR of PSAD and AR constructs probably display secondary structures that might mask the first AUG initiation codon. I think it would be reinforcing their claim if they bioinformatically analyzed these sequences to see if such structure may exist.

Reviewer 2 Report

In this manuscript, the authors describe the examination of several previously untested promoters of viral origin for transgene expression in C. reinhardtii. Even though the viruses belong to a lineage that can presumably infect Chlamydomonas, their tested promoters were not able to drive transgene expression in any reporter construct. While testing these promoters, the authors found out that the occurrence of alternative translations when they used mCherry as a reporter gene. They speculated that the 5’-UTR might affect this phenomenon and showed that it could be circumvented by the using other 5' UTR.

The manuscript is written in a clear and concise way, I just have a few comments about the content of the manuscript. See comments below for my questions about details.

Line 209: The picture of Ponceau S stains looks a bit weird and I can't really tell from them if the lanes were equally loaded.

Line 214: It would be nice to have the information about approximate number of colonies as well for the mCherry(i) reporter constructs.

Line 271: It is good to demonstrate that the replacement of the 5'-UTR works, but it would be even better to include additional bioinformatics analyses, such as secondary structure prediction. What are the predicted secondary structures look like? Does it conceal the first start codon? Is this also true for AβSAP(i)? Without this information, it is difficult to attribute the observed effects to the secondary structure of the 5'-UTRs in the abstract and discussion.

Fig3C: It would be good to add AβSAP(i)-mCherry(i) as well.

Discussion: It is unclear whether the differences in promoter strength measured by mCherry are due to the generation of alternative isoforms (does M10 have lower functionality than the original eGFP?), differences in mRNA levels, or weak translation. It would be helpful to compare the mRNA abundance, at least in the case of mCherry, if possible. Otherwise, at least it would be good to discuss.

Paragraph (line 302-314): There are many semicolons that can be replaced with commas.

Reviewer 3 Report

In the Methods, the transformation should be removed into a separate paragraph from the section strain and culture condition.

In Figure 1, what do the labeled Numerals besides 1 and 6?

Data analysis should be added in the Methods.

In Figure 2, what do the marks asterisk and dot mean? And what do the letters ns mean? What does the arrow head indicate?

Representative micrographs of the transformed cells with either bioluminescence and fluorescence signals or not are suggested to be supplied.

Round 2

Reviewer 2 Report

No further comments remain.